# Modeling Match Performance in Elite Volleyball Players: Importance of Jump Load and Strength Training Characteristics

**DOI:** 10.3390/s22207996

**Published:** 2022-10-20

**Authors:** Arie-Willem de Leeuw, Rick van Baar, Arno Knobbe, Stephan van der Zwaard

**Affiliations:** 1Department of Computer Science, University of Antwerp—IMEC, 2000 Antwerp, Belgium; 2The Dutch Volleyball Federation (Nevobo), 3528 BE Utrecht, The Netherlands; 3Leiden Institute of Advanced Computer Science (LIACS), Leiden University, 2333 CA Leiden, The Netherlands; 4Department of Human Movement Sciences, Faculty of Behavioral and Movement Sciences, Amsterdam Movement Sciences, Vrije Universiteit Amsterdam, 1081 BT Amsterdam, The Netherlands

**Keywords:** performance optimization, training load monitoring, volleyball, machine learning

## Abstract

In this study, we investigated the relationships between training load, perceived wellness and match performance in professional volleyball by applying the machine learning techniques XGBoost, random forest regression and subgroup discovery. Physical load data were obtained by manually logging all physical activities and using wearable sensors. Daily wellness of players was monitored using questionnaires. Match performance was derived from annotated actions by a video scout during matches. We identified conditions of predictor variables that related to attack and pass performance (*p* < 0.05). Better attack performance is related to heavy weights of lower-body strength training exercises in the preceding four weeks. However, worse attack performance is linked to large variations in weights of full-body strength training exercises, excessively heavy upper-body strength training, low jump heights and small variations in the number of high jumps in the four weeks prior to competition. Lower passing performance was associated with small variations in the number of high jumps in the preceding week and an excessive amount of high jumps performed, on average, in the two weeks prior to competition. Differences in findings with respect to passing and attack performance suggest that elite volleyball players can improve their performance if training schedules are adapted to the position of a player.

## 1. Introduction

Professional athletes aim for optimal performance in competition. Their performance depends on various factors, such as physical preparation and tactical decisions during the match. Which aspects have the largest impact on the performance depends on the sport that is considered. In volleyball, many studies have been conducted to investigate the relationship between a single performance indicator and match success [1]. For example, it is found that spiking and the number of block points are good indicators of match success [2] and that the effectiveness and type of attacks are important performance indicators for winning teams [3,4,5,6]. The main advantage of the aforementioned studies is that the outcomes can be used to optimize tactical preparation before competition [7].

In addition to tactical preparation, it is also important to physically prepare athletes by finetuning their training schemes. Scientific literature about the connection between physical preparation and match performance is scarce. In particular, the multivariate dependence of performance in competition on variables of training load has not been studied to date. Here, we aim to fill this gap in the literature by using a selection of machine learning regression techniques that have two important properties. First, we employ an explorative approach in which a variety of predictors is considered (while minimizing the risk of overfitting). As there is little scientific evidence as to exactly which predictors are important, this is a prerequisite for our technique of choice. For example, this is enables us to investigate the influence of predictors related to an athlete’s wellness, in addition to usually considered training load variables. This is potentially relevant because the mood or stress levels of an athlete before a match could impact their performance during the game. Second, our approach allows for potential non-linear relationships and possible interactions between predictors, which are important in the context of assessing injuries [8,9] and we expect to play a role here as well.

The aim of this study is to identify relationships between training load, wellness and match performance in elite volleyball players. As there are several positions on a volleyball team, different skills need to be optimized for different players. Because vertical jump height is an important indicator for attack effectiveness [10] and attacks blocked are important for winning or losing a match [11], we hypothesize that predictors related to high jumps are important for the performance of our action types. Finally, we will demonstrate that our results provide valuable insights for coaches that can be incorporated in training schedules.

## 2. Materials and Methods

### 2.1. Subjects

Twenty-five elite male volleyball players volunteered to participate in this study (mean ± SD age: 27.0 ± 3.0 years, weight: 91.2 ± 6.4 kg, height: 2.00 ± 0.10 m). All subjects competed on the international level, represented the same country and provided written informed consent. Only seventeen players competed in matches and had sufficient physical load data; therefore, eight of the players were excluded from our analyses.

Although data of only 17 players were included, data for multiple matches per player were analyzed. Performance was evaluated per action type, and only the most relevant actions for the player’s position were considered. A total of 122, 87, 58 and 35 data points were included for attacks, passes, blocks and services, respectively. Later, we will demonstrate that this number of records provides sufficient power to obtain statistically significant results from our machine learning analyses.

### 2.2. Experimental Design

Players were followed during their time on the national team during the 2018 international volleyball season. Over the course of 24 weeks, all training sessions and matches, as well as daily wellness, were monitored. The season consisted of two phases separated by 3 weeks of holidays during which the players had no training activities and no data were collected. The first period started with a 5-week preparation phase and was followed by 3 weeks of competition. After the holiday break, the players resumed training sessions for 10 weeks and finished this part with 3 weeks of competition.

The players participated in 6.1 ± 2.4 training sessions per week and trained for 13.8 ± 5.0 h per week. We excluded the holiday period without training activities. In total, the players competed in 31 matches, including 17 friendly matches.

### 2.3. Data Collection

In this study, we used several data sources. In Table 1, we briefly describe the most important groups of variables that are considered in this study. In the following sections, we will discuss each of these categories in more detail.

### 2.4. Match Performance

During every match, an experienced video scout annotated the actions of all players on the field using the volleyball-specific software program “*Data Volley 4*” (Data Project, Salerno (SA), Italy). This software is used by almost all international teams and allows coaches and analysts to convert volleyball actions performed on the court (attack, block, attack reception, freeball, pass, serve and set) into statistical data [1]. Attack reception is sometimes also referred to as defense. Based on rules and definitions [12], a rating is assigned to each action based on the execution. The video scouts use six distinct ratings (‘=’, ‘/’, ‘−’, ‘!’, ‘+’, ‘#’), which range from the best (‘#’ for actions that result in winning a point) to worst (‘=’ if the action leads to losing a point) possible rating. There is an exception for the serve, where ‘/’ is used for the second-highest rating. For this study, the same experienced video scout was responsible for the attribution of a rating for the volleyball performance of all actions during the matches. Results were double-checked to ensure that the annotation was consistent.

As an additional evaluation of the actions in a match, the Dutch Volleyball federation cooperated with the company ilionx to transform these ratings into importance scores for each action type based on historic data on international male volleyball matches between 2013 and 2019. For each action type and rating, the number of occurrences that led to winning a point was determined. After dividing this value by the total number of occurrences, every rating of each action type is transformed into a value between 0 and 10 based on the probability that this action will result in winning a rally. The final result of this scoring system is shown in Figure 1 and was used to generate a numeric performance target from the action ratings.

In our analyses, we focus on the actions that are most relevant for the position of a player. Offensive actions are the *serve* for a setter and the *attack* for outside hitters, middle blockers and opposites. Defensively, we consider the *blocks* for middle hitters, as well as *passes* for the libero and opposites.

### 2.5. Perceived Wellness

Every morning before breakfast (except for during the holiday break), players answered questions about their perceived wellness. Data were obtained with respect to fatigue, sleep quality, number of hours slept and mood. The answers ranged from 1 (very bad) to 10 (excellent) using a 10-point Likert scale [13,14]. At the beginning of this study, the players were already familiar with this questionnaire.

### 2.6. Training Load

For the matches and volleyball-specific training sessions, physical training load was recorded similarly. Here, the main focus is on jump characteristics. We collected the jump heights and total number of jumps using a G-VERT (Mayfonk Inc., Fort Lauderdale, FL, USA). The sensor is attached to the trunk using an elastic band, close to the center of mass [15,16]. The G-VERT measures 99% of all jumps higher than 15 cm, is in accordance with similar systems and has a high interdevice reliability [15,17].

Strength training sessions were also monitored in terms of the number of repetitions and sets. After completing an exercise, the players manually logged the number of repetitions and the applied weight in absolute kilograms. Therefore, possible changes with respect to the original strength training schemes were taken into account. Moreover, we accounted for interathlete differences by considering the applied weight as a percentage of the individual’s one-repetition maximum (1-RM).

The players used the CR10-scale [18] to assign a rating of perceived exertion (RPE) to all volleyball-specific and strength training sessions. By multiplying the duration of a session by its RPE score, we determined the session training loads. Based on these values, we obtained the training loads (summation of session training loads), monotony (day-to-day variation in training load) and training strain (overall training stress) [19].

### 2.7. Data Analysis

In every match, we determined the performance per action type of a player by averaging all ratings for this action type. For example, for an outside hitter, we recorded the average rating of all his attacks during a specific game.

In addition to quantifying match performance, we also needed to construct predictors that could potentially explain the variations in match performance. Because the literature provides little indication as to which characteristics of the variables are important for the match performance per action type, we needed to construct a broad collection of predictors. First, we constructed binary indicator variables for each player position by applying one-hot encoding. The construction of the predictor variables based on the perceived wellness or the internal and external training loads was more extensive, as follows.

We expected that the match performance would depend on the current physical condition of a player. As this physical condition is affected by all activities on preceding days, we performed multiple steps to construct the relevant predictor variables based on the perceived wellness and aspects of training load. Here, we followed the guidelines described in a previous study on overuse injury monitoring in elite volleyball players [20] and considered predictors that aggregate training load and wellness variables over specific time windows. First, we considered a variable that is measured over time (e.g., daily number of jumps). Then, we selected a certain time interval of interest prior to a match, such as one week, for which we considered this variable. Finally, we applied an aggregate function to construct a predictor variable. For example, one predictor variable is the average number of jumps per day in the week prior to a match.

In this study, we considered short-, mid- and long-term effects using time windows of the preceding 7, 14 or 28 days, respectively. Moreover, we considered the following 4 aggregate functions for all variables: first quartile, mean, third quartile and standard deviation. Additionally, the sum was considered for jump counts and weights of strength training exercises in absolute kilograms. Considering all combinations of these aggregate functions and window sizes, we derived multiple predictors from relatively few collected variables.

We constructed a total of 72 potential predictors from the number of jumps and jump height (as recorded by the G-VERT sensor). For jump counts we examine all jumps, as well as *low* (<50 cm), *average* (between 50 and 65 cm) and *high* (higher than 65 cm) jumps, separately. Moreover, we divided all strength training exercises into *full-body*, *lower-body* and *upper-body exercises* by considering whether the recruited muscles involved the full body, lower body or upper body, respectively. For example, a leg press is considered a lower-body exercise. We constructed 81 predictors that characterize the weight of the exercises either in absolute kilograms or in percentage of 1-RM. We constructed 48 predictors based on perceived wellness entries. We also determined 27 predictors for the training load, monotony and strain [19] and 9 predictors that specify the number of training sessions in different time windows. This resulted in a total of 237 predictors related to internal and external training load and wellness.

### 2.8. Machine Learning

In this study, we considered techniques that model all performance scores, in addition to discovering important predictors that distinguish between good and bad volleyball performance. Because the size of our data sets was small compared to the number of predictors, we had to limit the number of predictors included in our model to minimize the risk of overfitting. Therefore, applied machine learning regression techniques model volleyball performance by selecting only the most important predictors. Figure 2 shows a flow chart of our entire methodology.

### 2.9. Defensive and Offensive Performance Modeling

We started by applying the machine learning techniques called random forest [21] and XGBoost [22]. These techniques have multiple advantages, such as the flexibility to deal with different data types and the typically high accuracy of the models. Although we monitored all players on a team during the full 2018 international season, data were insufficient when constructing separate models per action type. Therefore, we categorized actions as offensive (serve and attack) and defensive (pass and block) actions. Then, we created models separately for all offensive and defensive performance scores.

Before we applied these machine learning techniques, for validation purposes, we applied stratified sampling on the target value and action types to make sure that the model was valid for low- and high-performance scores per action type and divided the data sets into a training and test set containing 70% and 30% of the data, respectively.

The training set was used to fit the machine learning models. Before training, our model, we first removed the features with near-zero variance. Then, we used 10-fold cross validation to tune the parameters (e.g., the number of trees) of the models. Subsequently, we constructed the model on the entire training set for the optimal parameter values and predicted either the offensive or defensive volleyball performance based on the predictor values for each data point in the test set. We then compared these predictions with the actual values and determined the predictive performance by calculating the mean absolute error (MAE) between predicted and actual values.

To assess whether the models had any predictive power, we also produced a *naïve baseline model* for both offensive and defensive performance. These models included no predictors and predicted the performance by taking the average of all performances in the training set. Therefore, the baseline models set an upper bound on the MAE, and only if the machine learning models had a significantly smaller MAE did the predictors explain part of the variance in the volleyball performance. We selected this measure over other options, such as the root mean squared error (RMSE), due to advantages in terms of interpretability. As the dependent variable (the offensive or defensive performance) is a numeric value between 0 and 10, the MAE corresponds to the absolute difference between the predicted and actual performance on a scale from 0 to 10. We also created an *action model* that uses only the action type as a predictor variable. In this case, the performance of a data point in the test set was compared to the average performance for the corresponding action type in the training set.

### 2.10. Predictors of Match Performance

To unravel the predictors that make a distinction between good and bad performance scores, we applied a third machine learning technique called *subgroup discovery.* This method has shown its potential in several sport settings by providing interpretable results that can easily be transferred to sport practice [23,24,25]. Moreover, the method is able to deal with small data sets.

In short, the method aims to find specific subsets of the data for which the dependent variable (in this case, volleyball performance) differs significantly from what is observed in the entire data collection. Here, the subgroup consists of all instances that satisfy a certain condition on the predictor variable(s), e.g., a threshold with respect to the number of jumps. With an automated search, the method investigates the effects of a multitude of conditions on the predictor variables. The significance of the results were assessed by running the algorithm on many randomized versions of the data. We determined the probability that an observed difference in terms of performance was a true finding or a false discovery by testing many hypotheses. A more detailed description of this method can be found in Ref. [20].

### 2.11. Subgroup Discovery Implementation

We applied subgroup discovery separately for all four action types. To avoid obtaining excessively specific results, we only considered subgroups with sizes between 10% and 90% of the size of the entire data collection. We considered the *z-score* to specify the difference between the subgroup and the entire data collection. Therefore, a positive or negative sign of the quality measure was considered to specify better or worse than average performance, respectively [26]. Finally, only subgroups that are described by a condition on a single predictor were investigated.

### 2.12. Statistical Analysis

The accuracy of our machine learning models for all defensive and offensive performance scores was determined by comparing the volleyball performance predictions of the models with the actual values for the test sets as represented by the mean and 95% confidence intervals (95% CI). The accuracies of the prediction models and the baseline models were compared using a paired sample *t*-test.

The significance of the subgroup discovery experimental results was assessed by performing a thousand runs on swap-randomized versions of the data and determining the probability that an observed difference in performance was a true finding or a false discovery by testing many hypotheses. We considered the results of our subgroup discovery experiments to be significant if the probability was less than 5% that the observed difference was a false discovery.

Finally, effect sizes were determined using Cohen’s *d* with corresponding 95% confidence intervals. We considered the effect sizes to be negligible (|*d*| < 0.20), small (0.20 ≤ |*d*| < 0.50), medium (0.50 ≤ |*d*| < 0.80) or large (|*d*| > 0.80) [27].

## 3. Results

### 3.1. Volleyball Performance

The distributions of overall match performance of the offensive and defensive actions are displayed in Figure 3. For the defensive actions, the block scores were 2.7 ± 1.5, and the pass scores were 8.6 ± 0.9. Thus, the average probability of scoring a point is higher for passes than for blocks. Moreover, the variation in the block scores is larger than for passes.

For the offensive actions, we reported attack scores of 6.8 ± 1.7 and service scores of 3.3 ± 0.9. In this case, the average probability of winning a point is higher for attacks than for services. Considering the variation with respect to the average scores, we found that the variation of attacks and services is comparable.

We also examined potential increases or decreases in performance over time using the Spearman’s rank correlation coefficient between match performance and match index within the season. The correlation coefficients were −0.02 (*p* = 0.19) and −0.19 (*p* = 0.02) for defensive and offensive performance, respectively. Thus, we observed no trend in the defensive performance throughout the season and a significant but negligible decrease in offensive performance over time.

### 3.2. Predicting Defensive and Offensive Match Performance Using Machine Learning Models

The accuracies of our machine learning models for all offensive and defensive performance scores are shown in Table 2. For both the offensive and defensive actions, we observed that machine learning models significantly outperformed the naïve baseline model. In total, the MAE was reduced by 36–47% and the 59–74% for offensive and defensive performance, respectively. Random forest was the best-performing model for offensive actions, and defensive actions were most accurately modeled by the XGBoost model. The hyperparameters of the random forest and XGBoost models are listed in Table 3. We found that even simple models that only distinguish between action types significantly outperformed the baseline model, as the average probability of winning a point differed depending on the action type.

### 3.3. Important Predictors of Match Performance Using Subgroup Discovery

The subgroups of our data collection with statistically differing performance for each action type are shown in Table 4. For block actions and services, we obtained no statistically significant results due to the limited sample size of 58 and 35, respectively. Furthermore, we obtained a single result for better attack performance and multiple subgroups for worse attack and passing scores.

Subgroup discovery analysis showed that variables related to strength training and jump load prior to the match are important predictors for attack performance. Specifically, the attack performance is better if the weights of 75% of all lower-body strength training exercises in the preceding 4 weeks are larger than 90 kg. Lower attack performance scores were associated with excessive strength training weights in the preceding 4 weeks. In particular, average weights of the upper-body exercises more than 0.9 percent greater than the 1-RM and a variation larger than 17.6 kg for the full-body exercises were related to worse scores for attacks. Moreover, we found that lower jump heights or insufficient variation in the number of high jumps (>65 cm) in the previous 4 weeks were related to lower attack performance.

For passes, variables related to high jumps were most important. Specifically, the performance was worse if the variation in the number of high jumps was larger than 9.75 jumps in the previous week or the average number of high jumps in the preceding two weeks was larger than 11.6.

## 4. Discussion

The aim of this study was to identify connections between training load, perceived wellness and volleyball performance. Here, we will discuss the results and mention the practical implications of our study.

### 4.1. Important Predictors

We found that the weights of lower-body strength training exercises in the 4 weeks prior to a match serve as a predictor for better attack performance. This could be explained by the fact that increased lower-body strength makes it easier for a player to achieve the acceleration necessary to jump high. Consequently, during the action, the player can focus more on other important aspects of attack performance, such as the appropriate ball placement, which could increase the chances of a successful attack. Furthermore, we found that lower jump height in the last 4 weeks is an important predictor of worse attacking performance. This is in agreement with the results of a previous study in which a positive correlation was found between vertical jump height and attack efficiency [10]. We also found that characteristics of strength training in the previous 4 weeks are relevant. In particular, the performance scores were worse if upper-body strength training weights were excessively heavy. A potential explanation could be that players lack precision and control of the ball if there is too much focus on increasing strength of upper body musculature.

For passes, predictors related to jumps were most relevant in our study population. High average jump heights in the week prior to a match were related to worse pass performance scores, which is the opposite relationship as that observed for attacks. Here, a heavy jump load prior to the match could have reduced the freshness of a player, potentially reducing their pass performance.

Previous studies on game performance in basketball demonstrate the importance of monitoring contextual factors in addition to the training load [28,29]. We also considered predictors related to perceived wellness and found that these predictors are not significantly related to performance in professional volleyball players.

In previous studies, no consistent dependencies were observed between external training load and match performance across different team sports [30]. In volleyball, most studies in this area focus on the physical performance of the players [31,32]. Only a few studies have used machine learning to investigate actual performance [33], with the main focus usually on technical and tactical aspects [34,35]. To the best of our knowledge, our study is the first to be conducted on elite volleyball players to investigate match performance using machine learning to examine the relationships between match performance and predictors based on training load and perceived wellness prior to competition.

### 4.2. Limitations

This study provides novel insights into the dependencies between volleyball performance and predictors based on training load and perceived wellness; however, this study is subject to some limitations. Most importantly, the aforementioned small number of matches prevented us from performing more detailed analyses to predict match performance.

To model the dependencies between all performance scores, training load and perceived wellness variables, we analyzed offensive and defensive actions separately. The result was that the best machine learning models performed almost just as well as a model based on average scores of the action types, mainly as a consequence of the observed difference in scores per action type, which was also previously observed in women’s volleyball [36]. This was also confirmed after determining the feature importance of predictors in the machine learning models [37]. For defensive performance, the player position is the dominating feature. When explaining the variance in offensive performance, jump load predictors are the most important predictor variables, but these also differ between the training programs per player position [38]. As we expect that the variables considered in this study and their interactions would be relevant for prediction of volleyball match performance, it would be interesting to collect more data points with respect to match performance for separate predictive modeling for each action type.

As previously shown in the context of overuse injury monitoring [20], we expect that personalization of training schedules has added value in terms of optimizing volleyball performance. Therefore, another interesting avenue for future research would be to perform player-specific analyses.

Finally, in our analyses, we solely focused on match performance. However, it could be worthwhile to also consider the match importance or the level of the opponent. For example, in future research, a weight factor can be introduced to distinguish friendly matches from matches during tournaments.

### 4.3. Practical Implication

Until now, it has been unknown which predictors might influence offensive and defensive volleyball performance. The added value of our machine learning approach is that it points to important predictors that may offer new directions for coaches and researchers to examine in terms of optimizing volleyball performance. In this study, the predictors related to training load explained differences in the attack and pass performance, but there were insufficient data to obtain statistically significant insights with respect to blocks and services. Therefore, our findings mainly have practical implications for players that focus on passing or attacks.

For example, we observe that lower-body and upper-body strength training in the 4 weeks prior to competition is an important predictor of worse attack performances. Therefore, coaches could slightly vary training schemes for offensive players to incorporate sufficiently heavy lower-body and less heavy upper-body strength training in this specific time period and observe whether and how this affects volleyball match performance. Moreover, based on our findings, it might be worthwhile to experiment with training schemes that restrict the variation in the number of high jumps in the week before a match or the average number of high jumps in the two weeks prior to a match for players focused on passing.

## 5. Conclusions

We used a machine learning approach to determine the dependencies between volleyball match performance and predictors related to training load and perceived wellness. We found that high weights of lower-body strength training in the four weeks prior to competition are related to better attack performance. Moreover, we identified important predictors related to worse performances in attacks and pass actions. In particular, excessively heavy upper-body strength training, large variations in weights of full-body strength training exercises, low jump heights and small variations in the number of large jumps in the four weeks prior to competition signal worse attack performances. Moreover, large variations in the number of high jumps (>65 cm) in the last week or an excessive number of high jumps in the previous two weeks indicate lower pass performances. Our findings can be used to personalize and finetune training schemes for individual players, thereby improving their performance in elite volleyball.

## Figures and Tables

**Figure 1 sensors-22-07996-f001:**
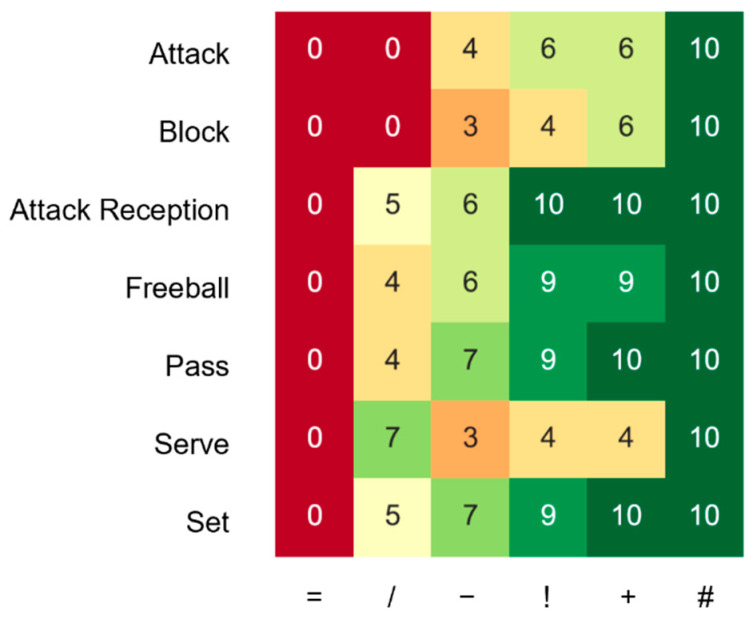
Overview of the rating of the volleyball actions. We distinguish seven action types and six ratings. Each combination of action type and rating is transformed into a score between the worst (0) and best (10) performance using the probability of winning a rally after performing an action with a certain rating based on data of male volleyball matches from 2013 to 2019.

**Figure 2 sensors-22-07996-f002:**
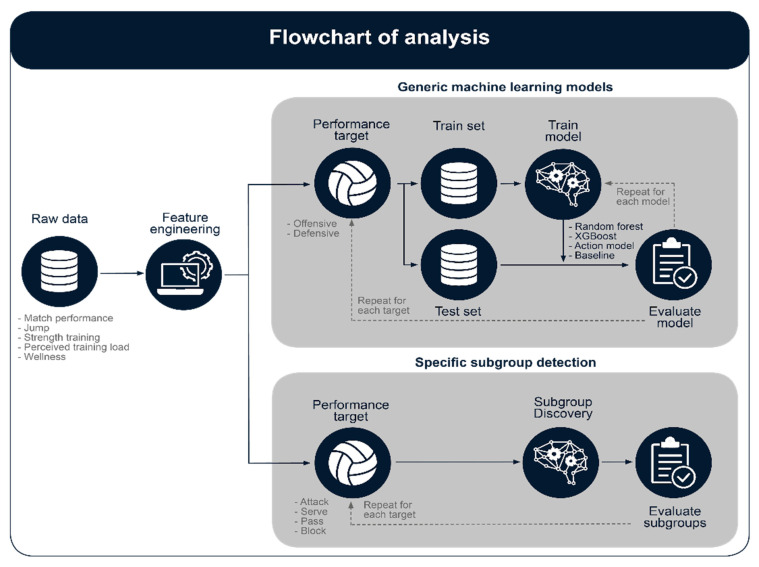
Flow chart of the analysis process for both the machine learning models and subgroup discovery analysis. Machine learning models were trained with hyperparameter tuning using 10-fold cross validation, whereas model performance was evaluated using the unseen test set. This analysis was repeated separately for offensive and defensive actions. Subgroup discovery results were evaluated with respect to the 5% false discovery probability (see Statistical Analysis). This analysis was repeated separately for attack, serve, block and pass actions.

**Figure 3 sensors-22-07996-f003:**
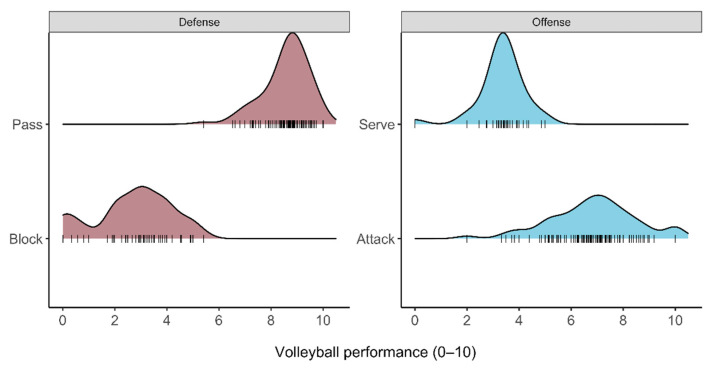
Distribution of scores for the considered volleyball actions. Defensively, the scores for pass actions are usually higher than the scores for block actions. Higher offensive scores are observed for attacks than for services.

**Table 1 sensors-22-07996-t001:** Overview of most relevant groups of variables.

Variable	Description
Match performance	Ratings of volleyball actions from 0 (worst) to 10 (best).
Jump load	Number of jumps and corresponding jump heights during matches and volleyball-specific training sessions.
Strength training	Weights of the executed exercises during strength training sessions in absolute kilograms or with respect to the 1-RM.
Perceived training load	Rating of perceived exertion (RPE) of all volleyball-specific and strength training sessions using the CR10-scale.
Wellness	Measures for perceived wellness on a 10-point Likert scale.

**Table 2 sensors-22-07996-t002:** Accuracy of the machine learning models for all offensive and defensive performance scores. Offensive performance is most accurately predicted by the random forest model, and XGBoost model is the best model for defensive actions. Models that only use action type as predictors are only slightly less accurate than our best machine learning models, suggesting that adding predictors related to external and internal training load does not result in large improvements in the prediction of offensive and defensive performance.

Action Type	Model	MAE (95% CI)	Difference in MAE	*p*-Value	Cohen’s *d* (95% CI)	Effect Size
Offense	Random Forest	0.91 (0.62–1.19)	−46.8%	*p* < 0.001	0.79 (0.47–1.18)	Medium
	XGBoost	1.09 (0.78–1.41)	−36.3%	*p* < 0.01	0.58 (0.23–0.99)	Medium
	Action Model	1.04 (0.75–1.32)	−39.2%	*p* < 0.001	0.66 (0.35–1.04)	Medium
	Baseline	1.71 (1.38–2.05)				
Defense	Random Forest	1.15 (0.79–1.51)	−59.4%	*p* < 0.001	1.47 (1.10–1.98)	Large
	XGBoost	0.75 (0.50–1.00)	−73.5%	*p* < 0.001	2.09 (1.63–2.78)	Large
	Action Model	0.79 (0.58–1.00)	−72.1%	*p* < 0.001	2.14 (1.63–2.88)	Large
	Baseline	2.83 (2.47–3.20)				

**Table 3 sensors-22-07996-t003:** Values of the hyperparameters for the random forest and XGBoost models for offense and defensive performances.

Model	Hyperparameter	Offense	Defense
Random Forest	ntrees	500	500
	mtry	10	10
	min.node.size splitrule	5 variance	5 variance
XGBoost	nrounds	50	50
	max_depth	1	1
	eta	0.3	0.3
	gamma	0	0
	colsample_bytree min_child_weight subsample	0.6 1 0.625	0.8 1 1

**Table 4 sensors-22-07996-t004:** Important features per action type revealed by the subgroup discovery analyses. We collected sufficient data to obtain significant results only for passes and attacks. The size of the subgroup is given as a percentage with respect to the size of the entire data collection for the corresponding action type. The description of the subgroups signals a condition on a predictor variable for which the performance scores are significantly different from all performance scores for the action type. A positive (negative) sign of the z-score corresponds to a better (worse) performance, with corresponding effect sizes determined by the absolute value of Cohen’s *d*.

Action Type	Description Subgroup	Z-Score Sign	Size	*p*-Value	Cohen’s *d* (95% CI)	Effect Size
Passes	Jumps_above65_ std7 ≥ 9.75	-	17.2%	*p* = 0.01	1.03 (0.48–1.62)	Large
	Jumps_above65_ avg14 ≥ 11.6	-	10.3%	*p* = 0.01	1.33 (0.64–2.08)	Large
Blocks	No significant results					
Serve	No significant results					
Attacks	LowerWeight_ firstquantile28 ≥ 90	+	15.6%	*p* = 0.01	0.83 (0.35–1.34)	Large
	JumpHeight_ thirdquantile28 ≤ 59	-	12.3%	*p* = 0.001	1.01 (0.47–1.58)	Large
	FullbodyWeight_ std28 ≥ 17.6	-	23.0%	*p* = 0.001	0.76 (0.35–1.19)	Medium
	WeightPrct_ Upperbody_ avg28 ≥ 0.90	-	28.7%	*p* = 0.001	0.67 (0.30–1.07)	Medium
	Jumps_above65_ std28 ≤ 2.24	-	10.7%	*p* = 0.02	0.99 (0.42–1.59)	Large
	WeightPrct_ Upperbody_ firstquantile28 ≥ 0.87	-	32.0%	*p* = 0.03	0.56 (0.20–0.94)	Medium

## Data Availability

The data presented in this study are available upon request from the corresponding author. The data are property of NeVoBo and therefore cannot be shared publicly.

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
