# Peer review of "Modeling Match Performance in Elite Volleyball Players: Importance of Jump Load and Strength Training Characteristics"

_sensors, 2022, doi:10.3390/s22207996_

Round 1

Reviewer 1 Report

The article presents an innovative approach, with methodological robustness but it has some weaknesses in terms of the theoretical framework that deserve improvement. The authors present a performance modeling in elite volleyball, using machine learning algorithms. For this, they intend to investigate the relationship between training load, perceived wellness and performance in competition.

The work presents refined methodological robustness that is not always accompanied by the proper theoretical/conceptual framework. In the introduction, for example, a brief reference is made to the potential influence of training loads on competitive performance (without developing it in detail), but the rationale for this study to analyze the athletes' perceived wellness data is not presented. Thus, it is recommended that the authors reformulate the introduction so that the analyzes carried out are supported by previous literature.

In the particular case of perceived wellness, the reason for considering these data in the study remains to be explained. It is only stated that "we obtained that these predictors are not significantly related to performance in professional volleyball players", however, this result is not properly discussed nor is the theoretical framework for implementing these variables presented. Defending that in science a "non-result" remains a result: I argue that these too should deserve analysis and discussion by the authors. In this particular case, I miss either a justification or a theoretical foundation for its inclusion in the study. Additionally, what is the basis for using a scale of 0 to 10 points, and not the more usual 0 to 7 points? It is recommended to include a reference to support this option.

The Materials and Methods section is the most complete and well-prepared. I congratulate the authors for the effort to present in detail the methodological procedures adopted in this work. I just miss a table with some descriptive data of the most relevant variables in this study, both performance and training load. I challenge the authors to build a table with basic descriptive statistics of the sample used in the machine learning models. 

I also recommend that there be an effort to standardize the technical concepts of volleyball throughout the article to avoid misinterpretation. In Figure 1, for example, the term "Defense" appears, but later this action is referred to as "pass" (line 434, for example). Or is this "pass" rather than a "set" (still using the terms in Figure 1)? 

Still in figure 1, what was the conceptual basis for the attribution of these ratings? Is there any supporting bibliography? Or was it a construction between the authors and the coaches of the observed team? Either option is valid, but clarification is recommended. 

Finally, it is important to clarify why a "Serve" of type "/" has a rating of 7 and theoretically better outcomes (such as "-", "!" and "+") obtain lower ratings. Was it a jackdaw? Or is it grounded?

Having tested different models, do the authors not consider it useful to show some metrics to evaluate them (in addition to the MAE)? It would be useful to know the confusion matrix, F1 or indicators of sensitivity, specifity, precision...

The discussion will also benefit from a restructuring: The "Machine learning and match performance" session is, in my opinion, complementary to the overall objective of the work and, if it arises, it should be at a later stage of the discussion. It seems to me that the "Important predictors" should be the central core of this discussion and deserve further development in the analysis and discussion of the results presented.

Overall, the approach presented in this work is extremely interesting and presents itself as a great framework for looking for relationships between performance indicators in sports performance and independent variables. I believe that if there is a better theoretical justification (with particular reference to the target variables of analysis) accompanied by a discussion focused on the results obtained (relationships found between performance variables and training indicators), this work has the potential to constitute an important milestone in this area of knowledge. In this sense, I encourage the authors to make the proposed changes aimed, above all, at increasing readability for future readers.

Reviewer 2 Report

This study investigates the relationships between training load, perceived wellness and match performance in professional volleyball by applying the machine learning techniques XGBoost, Random Forest regression and Subgroup Discovery. This is an interesting study. Some findings are reported, which can be incorporated in the training schedules for coaches. Some issues need to be addressed before publication.

1.     As most of the conclusions of this paper come from Subgroup Discovery analysis, what is the significance of using XGBoost and Random Forest regression algorithms to build models?

2.     This study mainly investigates the relationships between training load, perceived wellness and match performance. From Conclusions section, some qualitative results are obtained. Can the established XGBoost and Random Forest regression model be used for quantitative prediction?

3.     A flow chart of the analysis process of this article helps readers to quickly understand the main work of the article.

4.     This paper has obtained some findings through data analysis. If the author can explain the reasons behind these findings in essence (such as human function), the value of this paper will be significantly improved.

Round 2

Reviewer 1 Report

All my comments were sufficiently answered. In my opinion, the article presents itself as very competent to advance to the publication in sensors.